# Key Features of a Multi-Disciplinary Hospital-Based Rehabilitation Program for Children and Adolescents with Moderate to Severe Myalgic Encephalomyelitis/Chronic Fatigue Syndrome ME/CFS

**DOI:** 10.3390/ijerph192013608

**Published:** 2022-10-20

**Authors:** Sonya Hiremath, Montserrat Doukrou, Halina Flannery, Catherine Carey, Anna Gregorowski, Joseph Ward, Dougal Hargreaves, Terry Yvonne Segal

**Affiliations:** 1Victoria Hospital Kirkaldy, NHS Fife, Kirkcaldy KY2 5AH, UK; 2St Bernard’s Hospital, Gibraltar Health Authority, Gibraltar GX11 1AA, UK; 3University College London Hospitals NHS Foundation Trust, London NW1 2BU, UK; 4UCL GOS Institute of Child Health, London WC1N 1EH, UK; 5School of Public health, Imperial College London, London SW7 2AZ, UK

**Keywords:** adolescent, moderate, severe, chronic fatigue syndrome, myalgic encephalomyelitis, myalgic encephalopathy, outcomes, treatment, inpatient, measure, young people, long COVID

## Abstract

Purpose of the study: There is limited published data on treatment or outcomes of children and young people (CYP) with moderate or severe Myalgic Encephalomyelitis/Chronic Fatigue Syndrome (ME/CFS). Here, we describe outcomes of moderate and severe ME/CFS in CYP treated in a tertiary adolescent service. This information is useful when planning services for CYP and families affected by moderate/severe ME/CFS and to guide future management trials and commissioning decisions. Study Design: A retrospective review was conducted of medical records of the 27 CYP who received ward-based treatment in 2015. Notes were retrospectively reviewed to assess progress in four markers of wellbeing over the period of treatment: (i) mobility, (ii) education, (iii) sleep and (iv) involvement in social/recreational activities. Results: A total of 23/27 (85%) showed improvement in one or more domains over their period of ward-based therapy. 19/27 (70%) of patients showed improvement in physical ability. In 15/23 patients (65%), there was an improvement in ability to access education, in 12/24 (50%) sleep improved, and 16/27 (59%) demonstrated an improvement in socialising/ability perform recreational activities. Conclusion/Implications: A multidisciplinary hospital-based rehabilitation programme for moderate and severe ME/CFS was associated with improvement in at least one area of wellbeing in 85% of the CYP we reviewed. These data may be used as a baseline to evaluate the impact of other models of delivering care for this patient group. It may be useful when considering other groups such as those affected by Post-COVID Syndrome.

## 1. Introduction

Myalgic Encephalomyelitis (or encephalopathy)/Chronic Fatigue Syndrome (ME/CFS) is characterised by overwhelming, disabling fatigue with loss of physical and mental stamina, post-exertional malaise, sleep disturbance, significant reduction in usual activities and cognitive difficulties, among other symptoms [1]. A cardinal feature is post-exertional malaise, a usually delayed and often prolonged worsening of symptoms following physical or mental exertion. Diagnostic uncertainty and the complexity of the condition mean the prevalence of ME/CFS in children and young people (CYP) is unknown, with estimates varying from 0.1% to 4% [1,2,3]. ME/CFS in CYP can be characterised as mild, moderate, severe, or very severe (Table 1), with around 5–10% of patients thought to be severely affected [1]. Young people with severe/very severe ME/CFS are largely housebound and struggle with performing activities independently or can carry out minimal daily tasks only (such as face washing or cleaning teeth). They have severe cognitive difficulties, are often extremely sensitive to light and noise, and many depend on a wheelchair for mobility or are bedbound. They are often unable to leave the house and may have significant after-effects if they do so, and education is commonly disrupted [4]. This consequently makes normal social experience more challenging [2], and associated comorbid anxiety and low mood are common [4].

### 1.1. NICE Classification of Severity

Definitions of severity are not clear-cut because individual symptoms vary widely. The NICE definitions provide a guide on the level of impact of symptoms on everyday functioning. See Table 1.

### 1.2. Treatment

The NICE guideline for treatment of ME/CFS, published in October 2021, advises assessment and care and support planning should be delivered by an ME/CFS specialist team [5]. This includes a consideration of energy management, incorporating physical activity where appropriate, symptom management, sleep and rest, mobility, pain, orthostatic intolerance, diet, and emotional wellbeing, as well as relapse advice. The evidence in CYP for treatment is limited, although there is some evidence that cognitive behavioural therapy (CBT) may be beneficial [4,5,6]. Treatment is further complicated by the range of severity experienced by patients with ME/CFS. In severe cases, access to treatment is compromised by the condition itself, in terms of difficulty managing face-to-face appointments. More recently, the effects of the COVID-19 pandemic on healthcare delivery enabled us to offer virtual appointments between face-to-face admissions, making seamless care and support easier to deliver to moderately/severely affected patients [7].

Few research studies have been conducted in patients who are severely/very severely affected, and little is known about effective treatments in this patient group [1,2,7]. A recent small-scale study looking at a home-based, family-focused rehabilitative approach for severely affected adolescents with ME/CFS [8] demonstrated improved physical functioning and social adjustment after treatment, but no substantial improvements in fatigue in all participants.

### 1.3. Outcomes

The reported prognosis for ME/CFS is variable due to heterogeneity in methodology, such as case definition, length of time to follow-up and assessment of recovery. However, adolescents are more likely to improve and recover compared with adults [9]. Most paediatric studies report significant improvement or recovery in those treated; 54% to 94% improve within 5 years [10,11], whereas in adults, studies have shown <10% recovery to pre-morbid levels during the period of follow up [8]. There is very little research regarding effective management for severely/very severely affected CYP, and there is a need for further research in this area.

In view of the complexity of the condition and the spectrum of presentations, assessing improvement outcomes quantitatively is complex. Outcome measures used for this group often include fatigue scales such as the 11-item Chalder Fatigue questionnaire (CFQ), quality of life measures such as EQ-5D [12], the 36-item Short Form Survey (SF-36) or Paediatric Quality of Life (PedsQL) and/or measures of mood such as the Revised Children’s Anxiety and Depression Scale (RCADS) [13]. Pain scales such as the Visual Analogue Scale can be useful. In adults, there are significant ceiling effects and concerns with the reliability and validity of measures, such as between Multidimensional Fatigue Inventory-20 and RAND SF-36 subscales for ME/CFS adult patients [14]. There is limited evidence of the quality and acceptability of patient-reported outcome measures (PROMs) for CYP with ME/CFS.

### 1.4. Our Multidisciplinary Service

Our tertiary, nationally recognised service manages young people up to 19 years who are living with ME/CFS, chronic pain and persistent physical symptoms. The service offers a variety of tailored treatment options ranging from outpatient, coordinated one-day multidisciplinary team input (day case) to short inpatient stays (usually up to 1 week) within a teaching hospital. Moderately and severely affected patients access sets of regular 4–6 weekly coordinated day-case or inpatient admissions, where the CYP sees the medical team and allied health professionals following an individualised bespoke timetable. Reviews of treatment and progress are undertaken every 6 months. As one of the largest specialist adolescent centres for young people with ME/CFS, our data offer an opportunity to study outcomes among the moderate to severely affected group.

Within our service, occupational therapy, physiotherapy, specialist and ward nurses, and mental health and medical teams work closely together to provide bespoke, multi-faceted rehabilitation that CYP with such complex needs require. Input from inpatient education, safeguarding, and youth work is also included. Time is invested in building a trusting therapeutic relationship by hearing the patient’s journey and identifying what previous input has been helpful/unhelpful; acknowledging the symptoms experienced; understanding how these symptoms impact the lives of the CYP and family and appropriately reassuring patients that there is a way forward [15]. Goals are made collaboratively with CYP, ensuring opportunities to utilise their strengths and abilities. Where CYP have lost their sense of future, careful negotiations enable the patient to trust the therapists with setting the first goals until such a time as they are ready to take on a more active role. Time is spent helping CYP consider what is important to them and begin to build a sense of their preferred future, moving away from illness-dominated narratives of their lives and reconnecting with hopes, skills, and values. The range of treatments and techniques used are extensive due to the severity of symptoms and the significant impact these have on all aspects of patient functioning. Collaboration between the CYP, their family, the hospital team and the wider system, such as education and other medical professionals, is essential throughout every stage of the CYP’s journey.

## 2. Methods

We conducted a retrospective review of the medical records of CYP who received admissions (of a day or more) for multidisciplinary treatment in our adolescent unit over a year (2015). We included CYP aged 9–18 years at referral who met the NICE criteria for moderate and severe ME/CFS and were offered ward-based therapy in our service. We excluded CYP receiving exclusively outpatient therapy, and those admitted for medical admissions or without a diagnosis of ME/CFS. Some patients moved to inpatient input from outpatient management.

Although there is as yet no validated tool to assess outcomes in moderate to severe ME/CFS, health professionals working in this area have suggested the following areas [16] to be important to consider: symptoms, physical function; participation (activities including school and social life); and emotional wellbeing. These indicators have been used in previous work and overlap with those identified by CYP with ME/CFS as important in a recent study [17]. Based on this, we collected data on the following markers of wellbeing: (i) mobility/activity, (ii) education, (iii) sleep and (iv) involvement in social or recreational activities. Improvement in each indicator was defined as progressing across at least one degree of activity (Table 2).

Outcomes were measured by reviewing and scaling capabilities in these four areas as described by clinicians and therapists during their initial assessments in clinic and at discharge. Initial assessments ranged from 2011–2015 and notes were reviewed on discharge from the service (ranging from 2015–2018). The length of time CYP were in the service ranged from 10 months to 8 years, with an average of 2.8 years and median of 2.5 years. There was an average of 24 months and median of 27 months from the first admission to discharge. The year 2015 was chosen to ensure follow-up and discharge data by 2018 when the data were collected.

Criteria for reviewing discharge data were: (i) being discharged from the service to General Practitioner or adult ME/CFS services; or (ii) CYP and clinician decision to leave the service including those who had recovered. Data on demographic details and prior comorbidities were also collected. The results were then mapped graphically to capture how much each individual had improved, deteriorated or remained the same in each of the four categories upon leaving the service.

### Ethical Approval

This study was deemed exempt from formal NHS ethics approval by MRC/NHS HRA guidelines.

## 3. Results

In 2015, the service saw approximately 105 new patients and 415 patients followed up in clinic. We identified 27 CYP who met the inclusion criteria. A total of 18 (67%) were female, 8 (29%) were male and 1 (4%) was non-binary. The age at referral ranged from 9–17 years (median 15 years 6 months) and at discharge ranged from age 15–20 years (median 18 years 2 months). This cohort represents 5.2% of our patient population receiving treatment in our service, which reflects current estimates of prevalence of those severely affected by ME/CFS [1].

CYP were seen approximately every four weeks for one day (day case) (10 CYP) or 5-to-7-day admissions (8 CYP). Eight CYP had both day-case and inpatient admissions. The majority of CYP had between one and fifteen admissions (median 11–15 admissions). The most common co-morbidities in our cohort were anxiety, low mood, joint or muscle pain, and headaches. See Table 3 for the prevalence of and range of distribution of symptoms. Most CYP had more than one diagnosis and there were also patients with complex medical needs including functional neurological disorder and gastro-intestinal symptoms.

Specialist medical services were consulted about some patients with additional symptoms or concerns primarily to minimise over-investigation and provide further management (Table 4). Cardiology, Neurology and Gastroenterology were the services where the most support was utilised. Mental health practitioners were included in the service’s multidisciplinary offer.

Upon discharge from our service, 23/27 (85%) CYP showed improvement in one or more domains over their period of ward-based therapy. In total, 15 of the 23 (65%) CYP who were not in full-time education showed improvement. Overall, 12 of the 24 (50%) who had difficulties with sleep improved their sleep patterns during treatment. Further, 19/27 (70%) demonstrated improvement in physical ability and 16/27 (59.2%) showed improvement in their socialising abilities. Finally, 8/27 (30%) patients demonstrated improvement in all four areas.

### 3.1. Outcomes for Each Category

#### 3.1.1. Education

(i) Out of education; (ii) home tuition/online education; (iii) part time formal education; and (iv) full time formal education/work/enrolled in vocational/educational course. See Figure 1a and Table 5.

Of the 23 CYP who could show improvement in attending education/employment, 15 (65%) showed improvement upon leaving the service.

Of the seven CYP who started out of education, two went back to full time education with one enrolling in a bespoke college course allowing dual learning at home and at college while the other went on to attend college. Three were brought back into education with online or home tuition. Of the two patients who did not return to education or progress to work, one felt they had achieved some of their goals (improved mood and able to holiday). The other young person who remained out of education felt unable to engage with the service. They were unsure what they wanted from the service after six inpatient admissions, so they were in agreement to be discharged to local adult services.

All three who started home tuition/online education went back to part-time (1) or full-time (2) education.

Two patients went from part time formal education to out of education. Both of these patients had difficulties attending appointments for their therapies due to complex medical/mobility issues.

Of the three patients who went from part-time formal education to home tuition/online education, one patient completed their A-levels; therefore, this move was a positive outcome in actuality as they were able to continue their education whilst meeting their physical needs.

#### 3.1.2. Sleep

(i) A total of >12 h sleep per day/severe sleep phase delay/reversed pattern; (ii) difficulty getting up or getting to sleep/occasional napping; and (iii) routine sleep (8–10 h/night). See Figure 1b and Table 6.

Of the 24 patients with disordered sleep, 12 improved (50%), with a third of these gaining normal sleep patterns.

Four patients had severe sleep difficulties at the start and upon discharge. Of these, two patients felt they were still able to meet some of their personal goals and improved in other areas, so did not perceive it as a major difficulty for themselves. The other two patients were unable to engage with the service and were discharged with an open invitation to re-enter the service if they felt they needed.

#### 3.1.3. Physical Ability

(i) Bedbound; (ii) high level of dependence/wheelchair dependent; (iii) medium level of dependence or wheelchair-independent/able to walk short distances; (iv) low level of dependence/able to walk short distances and conduct activities of daily living/low-impact sports; and (v) able to walk as normal/take part in sports. See Figure 1c and Table 7.

Out of 27 CYP, 6 CYP achieved low-level dependence after treatment and 9 CYP were able to walk as normal compared to their peers and take part in sports if they wanted. A total of 19 of the 27 CYP (70%) improved their mobility during treatment, with 5 not showing any change and 3 decreasing in their level of ability.

Of the three patients who regressed, one went from low dependency to high dependency. Two patients went from high-level dependence to bedbound. In one this was due to increasingly difficult to manage complex medical problems, hence they were unable to attend the inpatient rehabilitation, and requested to be transferred to local services to continue their medical care closer to home. The other patient was discharged from the service due to being unable to engage with the service. These were the same two patients who struggled to improve in the other categories for the same reasons. The patient who regressed from medium-level dependence to high-level dependence had a multitude of concurrent medical conditions, requiring several medical appointments. In view of only being able to attend rehabilitation inpatient sessions when they had concurrent medical appointments, the patient and family felt they needed to be discharged from the service as they could not follow a consistent occupational therapy and physiotherapy programme.

### 3.2. Ability to Socialise/Recreational Activities

No = Not Socialising with friends, able to take part in their hobbies/interests,

Yes = Socialising with friends and/or able to take part in their hobbies/interests. See Figure 1d.

A total of 16 of 27 improved their socialising or participation in recreational activities.

## 4. Discussion

This study demonstrates how our multi-disciplinary approach with day case and admissions for intensive therapy effectively addresses the individual needs of CYP with moderate to severe ME/CFS.

We found that 23 out of 27 CYP with moderate/severe ME/CFS demonstrated improved mobility, sleep, educational attainment, or participation in recreational activities upon leaving our multidisciplinary hospital-based rehabilitation programme. Prior to our study, few data have been published on outcomes within CYP with moderate to severe ME/CFS who receive inpatient care. Results from our study provide a starting point that can be used to evaluate the impact and outcomes of new interventions or other models of delivering care for this patient group.

Using different metrics for improvement may have affected our results. For example, although 12 patients did not show a positive improvement in their sleep, the categories may have been too broad to detect an improvement. There may have been a mild improvement in hours of sleep gained that we have been unable to measure allowing the CYP to regain enough routine, enabling them to attend school, participate in activities and improve their physical activity level. It is also likely that improvement in other areas will gradually enable the CYP to improve their sleep habits as they have established routines elsewhere in their lives. Within education, our categories do not allow an improvement in hours to be identified unless the CYP has reached full-time attendance, so our results may underestimate improvement in this area. Our study looks for quantitative data as a method for outcome assessment, and this system does not showcase the personal goals for each young person, as we tailor our treatment aimed at patient goals.

There are different proposed phenotypes in ME/CFS and these are reflected in the cohort. Due to the small numbers of patients and the clinical heterogeneity, it was not possible to perform further statistical analyses nor allow interpretation of the reported results nor generalize them to other datasets or contexts.

It would be valuable to look in detail at those patients who achieved recovery (including full participation in education, physical activity, socialising and no physical dependence) and report on them separately.

Studying the multidisciplinary management of this group using larger cohorts and prospectively devised datasets may be useful going forwards.

Patient-reported outcome measures will be helpful in more accurately demonstrating meaningful progress. This in turn will help us to decipher whether any parts of the management are more effective than others.

The UK CFS/ME Research Collaborative report [18] identified the comparative lack of mainstream research funding into ME/CFS in the UK, highlighting a mismatch between the disease burden and the funding invested in research compared with other comparable physical illnesses. Further, although the new NICE ME/CFS guideline 2021 recommends the development of personalised care and support plans in collaboration with the person with ME/CFS and their family [5], the lack of guidance and research in this area may hinder the development of these plans. Other studies, such as the survey looking at gaps in treatments across European countries, support the development of these plans, and further work is needed in this area [19]. This particular study identified the need for incorporating individualised treatments and management in ME/CFS in order to achieve improvements in quality of life, especially in the severe/very severe ME/CFS patients. In addition, our study may be useful when considering management of other groups such as those affected by post-COVID Syndrome.

### Limitations

Limitations to our study include the use of an outcome framework that has not been validated.

Additionally, we acknowledge the lack of a control group in this study, and the potential influence of other factors that could influence outcomes.

CYP within our study were similar to those presenting to other specialist ME/CFS centres in the UK in terms of gender and symptomology, with pain, headaches and co-existing low mood being the most common features reported [20,21,22]. However, previous work has demonstrated that that whilst our local outpatient referrals broadly reflected the socioeconomic distribution, tertiary clinics showed a much higher proportion of patients from more affluent postcodes [23]. Further, our results may not be generalisable to other settings, such as primary care.

## 5. Conclusions

The tailored multidisciplinary programme described provides a starting point for personalised care and support plans, and most young people attending the service appear to improve in at least one domain during treatment. There could be other systemic factors influencing recovery.

The regular day case and admission model for intensive therapy may address the needs of CYP with moderate to severe ME/CFS.

Our tertiary multidisciplinary service treats CYP moderately and severely affected by ME/CFS with physical and psychological comorbidities. Following 1-to-7-day regular monthly admissions, for an average of 2.6 years, we were able to demonstrate an overall 85% improvement in one or more component of their mobility, education, recreation and sleep. In total, 19 out of 27 CYP (70%) improved their mobility, 12 out of 24 (50%) improved their sleep, 15 out of 23 (65%) improved in their educational/employment attainment and 16 out of 27 (60%) improved in their social abilities.

Further work is needed to define markers of positive outcomes, in order to assess the relative efficacy of different treatment options which will in turn impact service planning and commissioning of services. Patient-reported outcome measures will further complement our ability to measure treatment outcomes in this group of children and young people with moderate and severe ME/CFS.

### 5.1. What Is Already Known on This Topic

The estimated reported prognosis for young people living with moderate and severe ME/CFS is variable and uncertain. However, adolescents are more likely to improve and recover compared with adults.

There is little evidence on the best treatment in children and young people. There is a distinct lack of data on treatment and outcomes for the severely affected group.

### 5.2. What This Study Adds

Few research studies have been conducted in children and young people who are severely affected by ME/CFS, and little is known about effective treatments in this group.

This study describes the multidisciplinary model used by our team. It appears to identify positive outcomes for children and young people moderately and severely affected with ME/CFS treated in day care and inpatient rehabilitation settings.

Data from this study can be extrapolated to consider treatment options in other groups, such as those affected by Post-COVID Syndrome. See Table 8 for our key practice recommendations.

## Figures and Tables

**Figure 1 ijerph-19-13608-f001:**
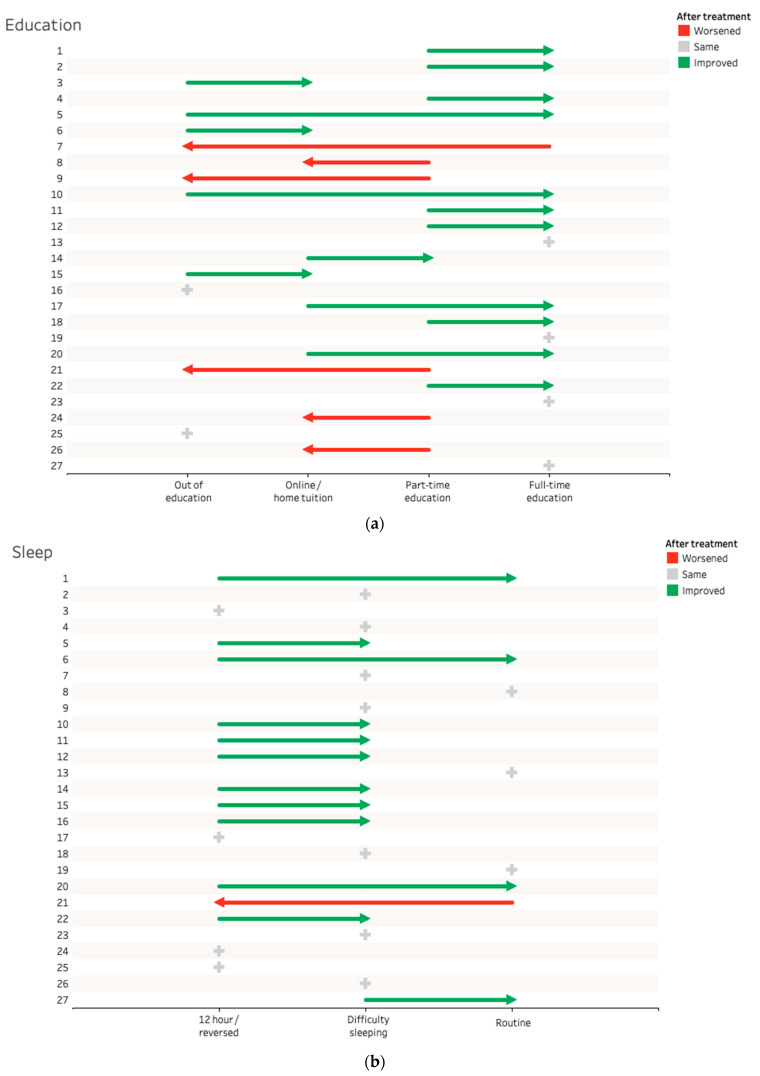
(**a**) Participation in education, (**b**) sleep, (**c**) participation in physical activity, (**d**) ability to socialise/take part in recreational activity.

**Table 1 ijerph-19-13608-t001:** NICE classification of severity [5].

**Mild ME/CFS**	People with mild ME/CFS care for themselves and perform some light domestic tasks (sometimes needing support) but may have difficulties with mobility. Most are still working or in education, but to do this they have probably stopped all leisure and social pursuits. They often have reduced hours, take days off and use the weekend to cope with the rest of the week.
**Moderate ME/CFS**	People with moderate ME/CFS have reduced mobility and are restricted in all activities of daily living, although they may have peaks and troughs in their level of symptoms and ability to do activities. They have usually stopped work or education and need rest periods, often resting in the afternoon for 1 or 2 h. Their sleep at night is generally poor quality and disturbed.
**Severe ME/CFS**	People with severe ME/CFS are unable to perform any activity for themselves or conduct minimal daily tasks only (such as face washing or cleaning teeth). They have severe cognitive difficulties and may depend on a wheelchair for mobility. They are often unable to leave the house or have severe and prolonged after-effects if they do so. They may also spend most of their time in bed and are often extremely sensitive to light and sound.
**Very severe ME/CFS**	People with very severe ME/CFS are in bed all day and dependent on care. They need help with personal hygiene and eating and are very sensitive to sensory stimuli. Some people may not be able to swallow and may need to be tube fed.

**Table 2 ijerph-19-13608-t002:** Outcome markers used.

Outcome Marker	Assessment Methodology
**Education**	Assessed by subdividing into 4 degrees of ability: Out of education Home tuition/online education Part time formal education Full time formal education/work/enrolled in vocational/educational course or apprenticeship.
**Sleep**	Assessed by subdividing into 3 degrees of ability: >12 h sleep per day or severe sleep phase delay/reversed pattern Difficulty getting up or getting to sleep/occasional napping Routine sleep (8–10 h/night).
**Physical Capability**	Assessed by subdividing into 5 degrees of ability: Bedbound High level dependence/wheelchair dependent Medium level dependence-wheelchair independent/able to walk short distances Low level dependence/able to walk short distances & conduct activities of daily living/low impact sports Able to walk as normal and/or take part in sports
**Social Capability**	Assessed by reviewing: Socialising with friends and/or ability to take part in their hobbies/interests

**Table 3 ijerph-19-13608-t003:** Prevalence of Symptoms in cohort.

Symptom	Prevalence (Number of Patients)
Fatigue	27
Headaches	14
Dizziness/feints	8
Sensitivity to light/noise	4
Joint/muscle pain	18
Abdominal pain	8
Constipation/diarrhoea	9
Nausea	6
Brain fog-reduced cognitive ability, impaired memory, reduced concentration)	3, 2, 5
Low mood, anxiety	12, 11
Weakness/paraesthesia	6
Urinary symptoms	5

**Table 4 ijerph-19-13608-t004:** Specialists involved.

Speciality Referred to	Number of Cases
Endocrine	5
Gastroenterology	9
Neurology	10
Pain team	6
Complementary medicine	4
Urology/gynaecology	7
Cardiology/autonomic unit	12
Rheumatology	4
Orthopaedics	4

**Table 5 ijerph-19-13608-t005:** Participation in education.

	On Admission to the Service	On Discharge from the Service
Out of education	7	5
Online/home tuition	3	3
Part time education	9	1
Full time education	4	15

**Table 6 ijerph-19-13608-t006:** Sleep.

	On Admission to the Service	On Discharge from the Service
>12 h or severe sleep phase delay	15	5
Difficulty getting to sleep or waking up or napping	8	15
Routine	4	7

**Table 7 ijerph-19-13608-t007:** Participation in physical activity.

	On Admission to the Service	On Discharge from the Service
Bedbound	1	2
High level dependence	3	1
Medium level dependence	12	5
Low level dependence	8	10
Normal (pre-morbid activity)	0	9

**Table 8 ijerph-19-13608-t008:** Key Practice Recommendations.

Key Practice Recommendations
A multidisciplinary approach can be helpful to assess and manage individual need in moderate and severe ME/CFS. Collaboration with children and young people living with ME/CFS and their families from the outset is advised. Developing an individualised care and support plan, in collaboration with the child, young person and family, designed around the child and young person’s preferred goals, is recommended in line with NICE guidance Offering virtual appointments in between face to face admissions can be beneficial. Building regular review into ongoing care and support is recommended

## Data Availability

Data provided in this study is not publicly available as due to small numbers in the study, patients may be identifyable.

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
