# Peer review of "Key Features of a Multi-Disciplinary Hospital-Based Rehabilitation Program for Children and Adolescents with Moderate to Severe Myalgic Encephalomyelitis/Chronic Fatigue Syndrome ME/CFS"

_ijerph, 2022, doi:10.3390/ijerph192013608_

Round 1

Reviewer 1 Report

Dear authors, I have some comments:

Lines 27 and 28, 190-191, 241-246 please check the font

Please check the reference number on the text and the references chapter according to the author’s guidelines

Table 1 -5 -maintain the same font in the table as in the text. The title for table 5 is missing.

Is this a typo? , teamv, line 58, beneficialivvvi, line 62, affectedvii line 67, groupiiivii 69, ME/CFSviii 71,

How did you differentiate depression as comorbidity from ME/CFS?

Reorganize the disproportionate discussions and conclusions.

There are other useful papers to be cited in this research:

Strand EB et al. (2019) Myalgic encephalomyelitis/chronic fatigue Syndrome (ME/CFS): Investigating care practices pointed out to disparities in diagnosis and treatment across European Union. PLoS ONE 14(12): e0225995. https://doi.org/10.1371/journal.pone.0225995

Aoun Sebaiti, M., Hainselin, M., Gounden, Y. et al. Systematic review and meta-analysis of cognitive impairment in myalgic encephalomyelitis/chronic fatigue syndrome (ME/CFS). Sci Rep 12, 2157 (2022). https://doi.org/10.1038/s41598-021-04764-w

Author Response

Dear Reviewer 1, Thank you for your helpful comments which we have aimed to address as described in the attached file.

Reviewer 2 Report

Valuable contribution; so little written on this topic

This paper represents a valuable contribution, given there are few research papers reporting on the management of ME/CFS in children and young people (CYP), particularly those with moderate to severe symptoms.  The paper is well written and the data are clearly presented. Suggestions for minor changes are provided to strengthen the paper and appropriately acknowledge the limitations of the study design. 

Methods:  Specifying the inclusion criteria would strengthen the paper, as would providing more detail on the patient population from which the study subjects were selected (at least the total number of patients).

In methods or results, please describe the distribution of length of follow-up. 

lines 137-144 : Adding details characterizing the sociodemographic characteristics and the distribution of comorbidities, diagnoses, and patterns of referral to specialists would be helpful.

Figures 1a-1d--the legends on the right side of these figures are not fully visible (outside margin)

Line 180:  It would be helpful to report separately on patients who achieved low-level dependence and no dependence, and to add denominators for the percentages.

Line 218:  A more rigorous study design including a control group would be needed to demonstrate effectiveness. Given its design, this study can only suggest that the day case and admissions for intensive therapy may address the needs of CYP with moderate to severe ME/CFS. For the same reasons, line 245 is somewhat of an overstatement.  A more accurate characterization would be that the program described provides a starting point for personalized care and support plans. In addition line 270 should be modified to avoid attributing the positive outcomes to the therapy program without qualification, given that there is no evidence presented in the paper to rule out the possibility that improvement might have occurred without therapy, or that factors other than the therapy might have influenced the outcomes.

Limitations: The lack of a control group and the fact that other factors that could influence the outcomes (apart from the intensive therapy provided) were not considered are important limitations that should be mentioned.

Key Practice Recommendations: This study did not provide sufficient evidence to support practice recommendations (see comments above re comparison group and other factors that might influence outcomes). Suggest changing the title to indicate instead that it describes the service, e.g. "Key features of a multi-disciplinary hospital-based rehabilitation program for children and adolescents with moderate to severe ME/CFS."

Reference citations were difficult to follow. Please correct errors in the numbering of citations throughout the paper and in the reference list. 

Author Response

Dear Reviewer 2

thank you for your helpful comments. We have attempted to address them and have made major revisions to the paper. The points are each addressed in the attached files.

Reviewer 3 Report

While the manuscript examines an important and timely topic, the lack of compelling statistical testing makes it hard for audience to interpret the reported results and generalize them to other datasets or contexts. Given the technical limitations and potential biases that are general confounds in any dataset, it is critical that rigorous statistical analyses are performed such that the reported observations can be properly interpreted, thereby minimizing the potential misinformation.

Author Response

Dear reviewer 3

thank you for your helpful comments. We have addressed these in the manuscript and added to the limitations. Please see the attached answer

Reviewer 4 Report

The manuscript covers an important topic to help improve the condition of young people suffering from ME/CFS. The collection of data is good, the results are interesting and should be brought to the public.

The main input is that the method, the results and the discussion should be better explained. Figure and table texts are missing and there is a need for more extensive explanations for the readers to understand what has been done and the results that are achieved.

The discussion part is very short whereas the limitation section is very long. It appears that some of the text in the limitation section should better be presented as part of the discussion.

There are several errors in the text on the references to the literature. For example, on page 3 line 69. Some references are in numbers, as in line 76, whereas most are in roman numbering. Roman numbering is very difficult to follow, and also the list of references in the literature list do not follow chronologically as they appear in the manuscript, by increasing number or alphabetically. I suggest using for example EndNote to handle the references.

Author Response

Dear Reviewer 4

thank you for your helpful comments. we have addressed them on the attached document

Round 2

Reviewer 1 Report

Dear authors, 

I appreciate your study of 22 moderate and severe ME/CFS in CYP, treated in a tertiary adolescent service, helping to plan services for CYP and families affected by moderate/severe ME/CFS and to guide future management trials and commissioning decisions. You could extrapolate your results to other clinical conditions, like post-COVIT exhaustion. 

Author Response

Please see file attached 

Reviewer 3 Report

My previous concerns have been addressed.

Author Response

Dear Reviewer 3

thank you for your comments. We are pleased that your previous concerns have been addressed with our revisions

Reviewer 4 Report

This is an important study that is now well presented.

Author Response

Dear Reviewer 4 

thank you very much for your supportive comments